# Visual Grounding with Multi-modal Conditional Adaptation

Ruilin Yao
Wuhan University of Technology
Wuhan, Hubei, China
Shanghai Artificial Intelligence Laboratory
Shanghai, China
yaoruilin@whut.edu.cn

Shengwu Xiong*
Wuhan University of Technology
Wuhan, Hubei, China
Sanya Science and Education Innovation Park
of Wuhan University of Technology
Sanya, Hainan, China
Shanghai Artificial Intelligence Laboratory
Shanghai, China
xiongsw@whut.edu.cn

Yichen Zhao
Sanya Science and Education Innovation Park
of Wuhan University of Technology
Sanya, Hainan, China
Wuhan University of Technology
Wuhan, Hubei, China
yichen_zhaoa@163.com

Yi Rong*
Wuhan University of Technology
Wuhan, Hubei, China
Sanya Science and Education Innovation Park
of Wuhan University of Technology
Sanya, Hainan, China
yrong@whut.edu.cn

## ABSTRACT

Visual grounding is the task of locating objects specified by natural language expressions. Existing methods extend generic object detection frameworks to tackle this task. They typically extract visual and textual features separately using independent visual and textual encoders, then fuse these features in a multi-modal decoder for final prediction. However, visual grounding presents unique challenges. It often involves locating objects with different text descriptions within the same image. Existing methods struggle with this task because the independent visual encoder produces identical visual features for the same image, limiting detection performance. Some recently approaches propose various language-guided visual encoders to address this issue, but they mostly rely solely on textual information and require sophisticated designs. In this paper, we introduce Multi-modal Conditional Adaptation (MMCA), which enables the visual encoder to adaptively update weights, directing its focus towards text-relevant regions. Specifically, we first integrate information from different modalities to obtain multi-modal embeddings. Then we utilize a set of weighting coefficients, which generated from the multimodal embeddings, to reorganize the weight update matrices and apply them to the visual encoder of the visual grounding model. Extensive experiments on four widely used datasets demonstrate that MMCA achieves significant improvements and state-of-the-art results. Ablation experiments further demonstrate the lightweight and efficiency of our method. Our source code is available at: https://github.com/Mr-Bigworth/MMCA.

## CCS CONCEPTS

• **Information systems → Multimedia and multimodal retrieval**; • **Computing methodologies → Object detection**.

## KEYWORDS

Visual Grounding; Vision and Language; Multi-modal Fusion

**ACM Reference Format:**
Ruilin Yao, Shengwu Xiong, Yichen Zhao, and Yi Rong. 2024. Visual Grounding with Multi-modal Conditional Adaptation. In *Proceedings of the 32nd ACM International Conference on Multimedia (MM '24), October 28-November 1, 2024, Melbourne, VIC, Australia.* ACM, New York, NY, USA, 10 pages. https://doi.org/10.1145/3664647.3681256

*Corresponding authors.

## 1 INTRODUCTION

Visual grounding aims to generalize traditional object detection to localization of regions in images that correspond to free-form text descriptions. Due to its potential in bridging the gap between visual perception and textual expressions, visual grounding have emerged as core problems in multi-modal reasoning [16, 19, 20, 50, 52].

Due to the similarity with detection tasks, early visual grounding approaches adhered to the established object detection frameworks, which evolved from initial two-stage approaches [24, 47, 49] to recently one-stage approaches [5, 23, 43]. Benifit from the transformer-based detectors DETR [2], TransVG [6] and MDETR [16] further propose an transformer-based framework which reformulate the prediction process to a end-to-end regression problem. Leveraging the excellent performance and scalability of transformers in multi-modal learning, these transformer based methods have achieved considerable results on visual grounding tasks. Nevertheless, the majority of these methods commonly adopt a sequential

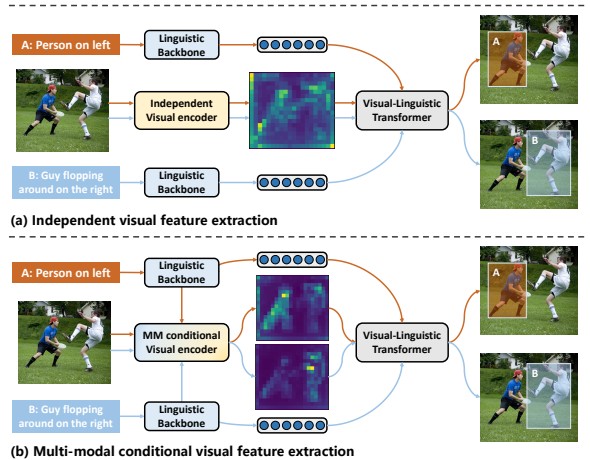

**(a) Independent visual feature extraction**

**(b) Multi-modal conditional visual feature extraction**

**Figure 1: (a) Traditional visual grounding framework with independent visual encoder. (b) Our proposed visual grounding framework with Multi-modal (MM) conditional visual encoder. We visualize the ground truth and the attention maps of various visual encoders. The attention distribution of the independent visual encoder appears more diffuse, whereas the attention distributions of the MM-conditional visual encoder are more concentrated on the corresponding object.**

extraction and fusion approach. This involves employing independent visual and textual encoders to extract feature, which are subsequently input into a multi-modal decoder to generate prediction results. However, the absence of interaction between modalities constrains the performance of detectors in visual grounding tasks.

For visual grounding tasks, the role of the visual encoder is to extract potential foreground features guided by the prior knowledge acquired during training. Due to the fact that the same image often corresponds to multiple different objects associated with unique textual expressions, the independent visual feature extraction process limiting the visual encoder, which can only trained to extract compromised general visual feature rather than textually specific one. As illustrated in Figure 1 (a), the attention map of the independent visual encoder highlights general salient regions but struggles to focus on the most text-relevant regions, which result in the gap between the visual feature and the feature required in multi-model reasoning. Consequently, the independent modality-specific encoder fails to fully adapt to the requirements of visual grounding.

Several previous works have noticed this problem, VG-LAW [34] proposes a language-guided dynamic visual network by generating weights for visual encoders using textual information. LADS [33] employs a language-guided gating mechanism to achieve dynamic inference of visual input. These methods dynamically modify the parameters or the architecture of the visual encoder according to the textual features, as illustrated in Figure 2 (a). Some other methods, such as QRNet [46] and LAVT [44], improve the post-fusion paradigm for visual and textual features through integrating linguistic information into visual feature at intermediate levels based on elaborate attention modules or additional feature-adjustment modules, as shown in Figure 2 (b). While these methods achieve

appreciable performance, most of them still require sophisticated designs, such as language-guided attention modules [44, 46], complex weight generation process [34], or gumbel-softmax technique[33]. Additionally, all above methods rely solely on textual information, which may limit flexibility in certain applications and be susceptible to the quality of the expressions.

In this paper, we aim to explore an efficient and lightweight interaction strategy from the perspective of transfer learning. Inspired by the efficiency of LoRA [14] in adapting to different downstream tasks, we propose the Multi-modal Conditional Adaption (MMCA) to guide the visual encoder to focus on the text-relevant regions, as depicted in Figure 2 (c) and (d). We consider language-guided visual feature extraction as a downstream task of general visual feature extraction, and regard the process of adapting visual encoders to different expressions as a weight update process relying on multi-modal information. Specifically, visual and textual features are integrated through a gating mechanism to obtain multimodal embeddings, and multi-modal conditional adaptation involves a set of low-rank weight matrices reorganized from the coefficients generated by these multimodal embeddings. During inference, the visual encoder can adaptively update its weights through these matrices. Thus, for a given image input, the visual encoder can focus more on the foreground regions associated with the expression, as Figure 1 (b) shows. We benchmark our proposed method based on TransVG [6] on four prevalent datasets, including RefCOCO [48], RefCOCO+ [48], RefCOCOg [28], ReferItGame [18] and our method achieves comparable results with the state-of-the-art methods. Furthermore, when applying MMCA to various stronger baseline models, it consistently brings consistent improvements. Ablation studies also compare the performance of different variants of our proposed method and report the parameters and the inference speed, revealing that our approach is lightweight and efficient. In summary, we make three-fold contributions:

- We propose the Multi-modal Conditional Adaption (MMCA) method, which improving the feature extraction process of the visual encoder in the visual grounding model from a novel weight update perspective.
- We apply the proposed MMCA to the mainstream visual grounding framework and propose the flexible multi-modal conditional transformer and convolution module, which can be easily applied to other visual grounding models as a plug-and-play component.
- We conduct extensive experiments to verify the effectiveness of our method, and the results on four representative datasets showcase a significant improvement with a small cost.

## 2 RELATED WORK

### 2.1 Visual Grounding

Visual grounding aims to ground a natural language description onto the referred region in an image. Due to inheriting the general object detection framework, early visual grounding methods can be broadly categorized into two directions, i.e., two-stage methods [24, 41, 47, 49] and one-stage methods [5, 23, 43]. Two-stage methods match the language feature to the vision content at the region level, thus requiring the vision encoder to first generate a set of region proposals. One-stage methods densely perform multi-modal feature

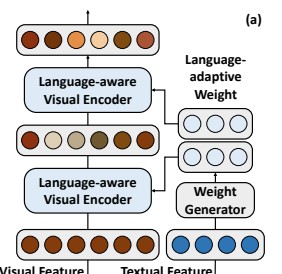 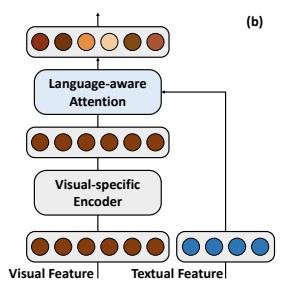 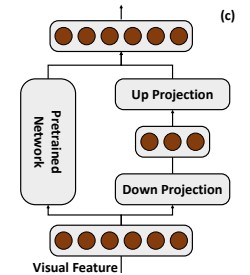 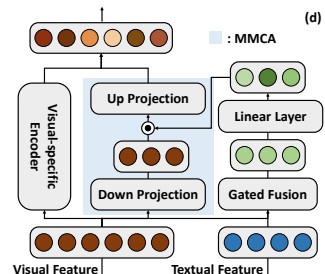

**Figure 2: (a) The parameters or the inference pipeline of the visual encoder are dynamically modified according to the textual feature. (b) Integrating textual and visual features through finely designed attention modules. (c) LoRA uses the additional trainable low-rank parameter matrices to simulate weight updates in transfer learning. (d) MMCA utilizes multi-modal information to control a set of update matrices for the visual encoder to realize language-guided visual feature extraction.**

fusion at all spatial locations, waiving the requirements of region proposals, and predict the location of referred object directly.

With the success of transformer in detection and vision-language tasks, a series of transformer applied to visual grounding tasks have been proposed. Referring Transformer [21] leverages contextualized phrase queries and directly decodes them into corresponding image regions and segments. TransVG [6] incorporates DETR encoder [2] to extract visual feature and proposes a multi-modal reasoning module. MDETR [16] directly predicts the bounding boxes of the objects by a transformer encoder-decoder which use the aligned modulated feature as the input. Although transformer-based methods [6, 21] achieve better performance in visual grounding tasks benefiting from the self-attention mechanism. The independent visual encoder, may difficult to focus on the text-relevant regions, limits its performance for visual grounding tasks.

## 2.2 Parameter-Efficient Transfer Learning

Transfer learning aims to produce the fine-tuned model, which adapts to the specific task or dataset, based on the pre-trained model (either via the supervised or the unsupervised manner). Transferring the large pre-trained models [1, 7] into downstream tasks has been the popular paradigm for a long time. Conventional arts [1, 7, 26] training all the network parameters to make them adapt to the target tasks. However, with the growth of model sizes and the complexity of the specific tasks, the full-parameter fine-tuning paradigm is inevitably limited by the huge computational burden and catastrophic forgetting.

To alleviate these issues, some parameter-efficient fine-tuning methods have been proposed. One approach, known as Prompt Tuning [10, 15], addresses the distribution mismatch between pre-training and downstream tasks by learning task-specific tokens. Adapter-like methods [4, 13, 17] insert trainable modules, such as MLPs with activation functions and residual structures, into the network to facilitate transfer learning. LoRA-like methods [14, 51] exploit the low-rank update to a large-scale frozen model and introduces a bypass to the original parameter matrix to mimic the fine-tuning of the entire model parameters. Inspired by the success of NLP, several notable works [4, 35, 39] have emerged in the computer vision domain. And they provide an efficient way to adapt a model to specific tasks, inspiring us to improve visual

grounding tasks by adaptively update the weight of the visual encoder through text guidance.

## 3 METHODS

We focus on the challenge of language-guided visual feature extraction in visual grounding tasks. We introduce the architecture of our method in the initial section. In the subsequent section, we outline how multimodal information is utilized to guide the feature extraction process of the visual encoder through weight updates, with the objective of emphasizing regions pertinent to specific expressions. And we expound upon our approach to integrating visual and textual features, which aims to mitigate the influence of potential low-quality text on language-guided visual encoders. Finally, we show how to apply our method to visual grounding model and propose the multi-modal conditional transformer and multi-modal conditional convolution module. The overall pipeline of our model is schematically illustrated in Figure 3.

## 3.1 Overall Architecture

Here we present the architecture of the adopted visual grounding framework, which follows the typical end-to-end encoder-decoder paradigm [6]. Illustrated in Figure 3, given an image and a language expression as inputs, we initially feed them into the encoder part to generate corresponding feature embeddings. In the Linguistic Branch, the linguistic backbone take the tokenized language expression as input and extract the textual features $f_t \in \mathbb{R}^{N_t \times C_t}$, where $N_t$ is the number of language tokens. Meanwhile, in the Visual Branch, a CNN backbone first extracts a 2D feature map, followed by a stack of transformer encoder layers that generate a flattened sequence of visual features $f_v \in \mathbb{R}^{N_v \times C_v}$. Our proposed Multi-modal Conditional Adaption (MMCA) is hierarchically applied to the parameter matrices of the convolutional and transformer layers. This module takes both visual and textual features as inputs and dynamically updates the weights of the visual encoder to achieve language-guided visual feature extraction. Subsequently, we concatenate the visual and textual feature embeddings and appending a learnable token, [REG] token, as the inputs for the multi-modal decoder (Visual-Linguistic Transformer), which embeds the input tokens from different modalities into a aligned semantic space and perform intra- and inter-modal reasoning with the self-attention

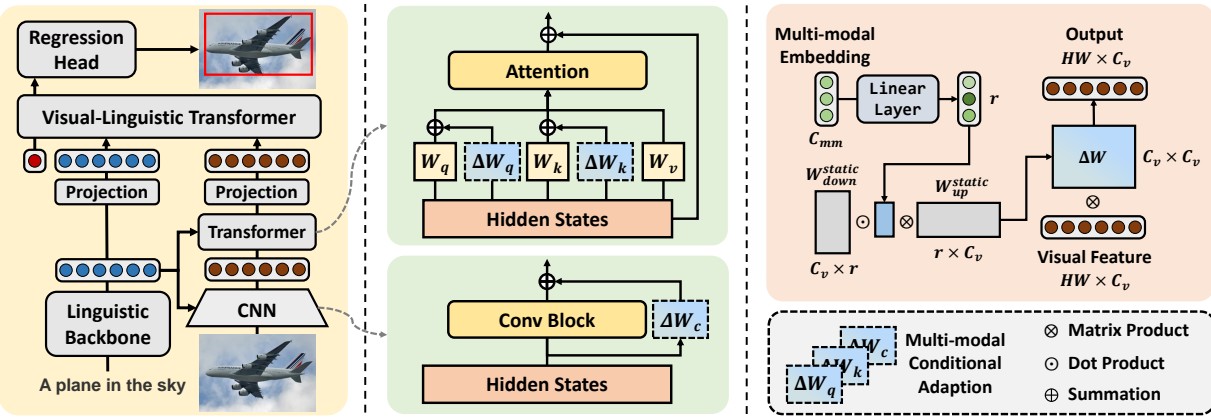

**Figure 3: Overview of our proposed Multi-modal Conditional Adaption framework. We obtain a multi-modal embedding from visual and textual features and input it into different layers of the visual encoder to reorganize a set of weight update for the visual encoder. The figure shows the conditional weight update for the self-attention layer (query and key) and convolution layer in the visual transformer and CNN backbone.**

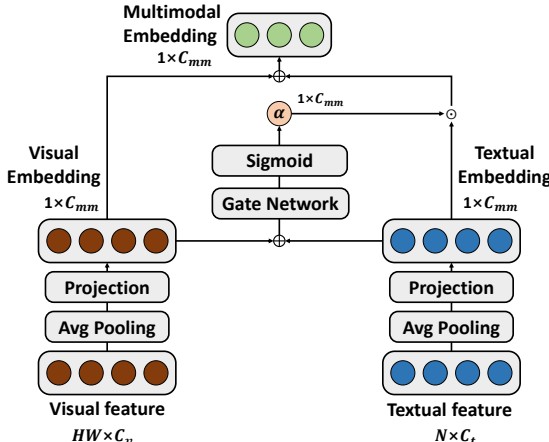

**Figure 4: The gated fusion of visual and textual features.**

layers. Finally, the regression head uses the output state of [REG] token to directly predict the 4-dim coordinates $\hat{b} = (\hat{x}, \hat{y}, \hat{w}, \hat{h})$ of the referred object. The training loss with the ground-truth box $b = (x, y, w, h)$ can be formulated as:

$$\mathcal{L} = \mathcal{L}_{smooth-l1}(\hat{b}, b) + L_{giou}(\hat{b}, b) \qquad (1)$$

where $L_{smooth-l1}(\cdot)$ and $L_{giou}(\cdot)$ are the smooth L1 loss [11] and GIoU loss [31], respectively.

## 3.2 Multi-modal Conditional Adaption

The existing methods employ various strategies for visual feature extraction with language guidance. Although performance gains can be achieved with these methods, most of them encounter the challenge of requiring sophisticated designs and relying solely on textual information, which can be susceptible to the quality of the referring expression, especially in complex scenes. To address these challenges, our method integrates visual and textual information to obtain multimodal embeddings. And we use these embeddings to guide the visual encoder in a weight updating manner, allowing

the model to adapt to various referring expressions directly. In the following, we will first introduce the implementation of conditional adaptation for visual grounding tasks. Then, we will detail the generation of multimodal embeddings via gated fusion for the visual and language inputs.

**Conditional Adaption.** In order to efficiently adapt the pre-trained model to downstream tasks, LoRA [14] models the incremental update of the pre-trained weight matrix, typically performed by the dense layers in the neural network, by the product of two low-rank matrices. For $h = Wx$ where output $h \in \mathbb{R}^d$, input $x \in \mathbb{R}^k$ and weight matrix $W \in \mathbb{R}^{d \times k}$. When adapting to a specific task in standard transfer learning, LoRA hypothesizes the update to the weights matrices have a low "intrinsic rank" during adaptation and modified forward pass yields:

$$h = W_0 x + \Delta W x = W_0 x + BAx \qquad (2)$$

where $W_0 \in \mathbb{R}^{d \times k}$ represents the pre-trained weight matrix, $\Delta W$ denotes the weight update and $B \in \mathbb{R}^{d \times r}, A \in \mathbb{R}^{r \times k}$ with $r \ll min(d, k)$ are the low-rank adaptation based on the "instrisic dimension" assumption. Benefits from LoRA, a pre-trained model can be shared and efficiently switch to different downstream tasks by update a small number of parameters.

For the visual grounding task, we hope that different referring expressions can control a set of weight updates for the visual encoder, thereby directing the encoder's focus towards text-relevant regions. While directly generating such matrices brings about two drawbacks. (1) It requires a large parameter generator, i.e., using a linear projection to generate the matrices $B \in \mathbb{R}^{d \times r}, A \in \mathbb{R}^{r \times k}$ from a embedding $E \in \mathbb{R}^{d_{mm}}$ needs $(d_{mm} + 1) \cdot (d \times r + r \times k)$ parameters. (2) The generator without constraints may overfit the expressions in training, while hardly understand the expressions during testing [46]. Motivated by some previous works [9, 37, 38] which empirically verified that the interpolation in weight space can maintain the model robustness for data from different distribution. We enable the network to learn a set of basis matrices of the weight update and use multi-modal information to reorganize the

update matrices, as shown in Figure 3, which allows the parameter generator to be lightweight and ensure the weights of the network are updated in the same space.

Specifically, we first decomposing the weight update matrices and reformulate it as a sum of outer products:

$$\Delta W x = BAx = \sum_{i=1}^{r} B_i \otimes A_i \quad (3)$$

where $B_i, A_i \in \mathbb{R}^{d \times 1}$ represents the $i-th$ column and row of $B, A$. Then we keep the form of out products by $B_i, A_i$ and use a weighted sum to control the subspace of the adaptation:

$$h = W_0 x + \Delta W x = W_0 x + \sum_{i=1}^{r} w_i B_i \otimes A_i \quad (4)$$

here $W_0 \in \mathbb{R}^{d \times k}$ denotes the fixed, independent weight matrix of visual encoder, $\Delta W$ denotes the conditional weight update, $B \in \mathbb{R}^{d \times r}, A \in \mathbb{R}^{r \times k}$ here are static low-rank weight matrices and the coefficients $w_1, w_2, ..., w_r$ are scalar generated from multi-modal embedding. For simplicity and not introduce other inductive bias, we use a linear regression to generate this set of weights:

$$[w_1, w_2, ..., w_r]^T = W_g E_{mm} + [b_1, b_2, ..., b_r]^T \quad (5)$$

where $W_g \in \mathbb{R}^{r \times d}$, $[b_1, b_2, ..., b_r]^T$ are parameter matrices and $E_{mm} \in \mathbb{R}^d$ is the layer-specific multi-modal embedding, generated from the textual features and the visual features output from the previous layer. Unlike transfer learning tasks, we do not aim to fine-tune a little part of parameters to adapt the specific downstream task, but rather hope that the visual encoder can adapt various expressions. So all parameter matrices $W_0, B, A, W_g, [b_1, b_2, ..., b_r]^T$ are learnable during the training phase.

**Gated Fusion for Multi-modal embedding.** As previously discussed, relying solely on textual information to guide visual encoders may restrict flexibility in certain applications, and performance may be impacted by the quality of the textual information. To mitigate these issues, we employ gating mechanisms to regulate the input of textual information. Given the textual features $F_t \in \mathbb{R}^{N_t \times C_t}$ and the flattened visual feature $F_v \in \mathbb{R}^{HW \times C_v}$, we first apply pooling operations to process textual features of different lengths and visual features of different spatial sizes. Subsequently, linear projections are used to generate fixed-dimensional embeddings $E_t, E_v$ for the respective modal-specific features. We then employ a simple gating mechanism with a sigmoid activation to fuse the visual and textual embeddings:

$$E_t = W_t F_t, E_v = W_v F_v \quad (6)$$

$$\alpha = \sigma[W_g^1 \delta(W_g^2 (E_t + E_v))] \quad (7)$$

where $\delta$ denotes ReLU, $W_t \in \mathbb{R}^{C_{mm} \times C_t}$, $W_v \in \mathbb{R}^{C_{mm} \times C_v}$, $W_g^1 \in \mathbb{R}^{C_{mm} \times \frac{C_{mm}}{k}}$ and $W_g^2 \in \mathbb{R}^{\frac{C_{mm}}{k} \times C_{mm}}$ are trainable parameter matrices. $\alpha \in \mathbb{R}^{C_{mm}}$ controls how much textual information is kept. To limit model complexity and aid generalisation, we parameterise the gate mechanism by forming a gate network with two fully-connected (FC) layers around the non-linearity. And the output of gated fusion is obtained by summing the visual embedding with the rescaled textual embedding:

$$E_{mm} = \alpha E_t + E_v \quad (8)$$

Finally, the fusion embedding $E_{mm}$ is utilized to generate the coefficients, which guiding the weight update for visual encoder.

### 3.3 Applied for visual grounding

Finally, we show how to apply our method to the adopted visual grounding model. Based on the visual encoder (Convolutional and Transformer Layers), we then propose the Multi-modal conditional transformer and Multi-modal conditional convolution by applying the proposed MMCA as follows:

**Multi-modal Conditional Transformer.** The transformer encoder layer in visual backbone mainly consists of two types of sub-layers, i.e., MHSA and FFN. In MHSA, the visual feature $X$ are linearly projected by embedding $W_q, W_k$ and $W_v$ into three vector. And the output of the MHSA is performed on these vectors by:

$$h = softmax(\frac{W_q X X^T W_k^T}{\sqrt{d_k}}) W_v X + X \quad (9)$$

where $h$ are the tokens produced by MHSA. In FFN, the output tokens are further sent to a LayerNorm and a MLP block which is consisted of two fully connected layers with a relu activation in between. This process is formally formulated as follows:

$$X_{output} = LN(MLP(h') + h') \quad (10)$$

where $X_{output}$ is the output of the transformer encoder block and $h' = LN(h)$. By applying Multi-modal Conditional Adaption, our method can be represented as:

$$h' = softmax(\frac{(W_q') X X^T (W_k'^T)}{\sqrt{d_k}}) W_v X + X \\ W_q' = W_q + \Delta W_q, W_k' = W_k + \Delta W_k \quad (11)$$

$$X_{output} = LN(MLP(h') + \Delta W_m h' + h') \quad (12)$$

where $\Delta W_q, \Delta W_k, \Delta W_m$ are the conditional weight update for the linear projection of query, key and MLP block. It is noted that we take the embedding $\Delta W_q, \Delta W_k$ as an example in Figure 3 and we would discuss the impact of applying our method to different type of weight during ablation study.

**Multi-modal Conditional Convolution.** Consider the commonly used convolution block $Conv_{k \times k}$ with a weight update denoted as $\Delta W_c \in \mathbb{R}^{c_{in} \times c_{out} \times k \times k}$, where $k$ represents kernel size, $c_{in}$ and $c_{out}$ indicate the number of input channels and output channels. To facilitate the application of our method, we unroll this weight update into a 2-D matrix represented as $\Delta W_c \in \mathbb{R}^{c_{out} \times c_{in} k^2}$ and approximate this update with two matrices $B \in \mathbb{R}^{c_{in} \times r}, A \in \mathbb{R}^{r \times c_{out} k^2}$ with rank $r$. Such a decomposition implies that the given weight update can be approximated by two consecutive convolutional layers $Conv_B$ and $Conv_A$ with kernel sizes 1 and $k$ which use $r$ as output and input channels. With the preceding analysis, the Multi-modal Conditional Adaption for convolution block can be expressed as:

$$X_{output} = Conv_{k \times k}(X) + Conv_A(W_{mm} \odot Conv_B(X)) \quad (13)$$

where $X$ and $W_{mm} = [w_1, w_2, ..., w_r]^T$ are the visual feature from the previous convolutional layer and the weighting coefficients generated from the multi-modal embedding. We perform dot product between the coefficients and the output of $Conv_B$ along channel dimension and feed the output into $Conv_A$, which is equivalent to reorganize the weight update. Through this process, we achieve the multi-modal conditional convolution. For the convolutional visual backbone (ResNet), we treat the bottleneck block as a independent

**Table 1: Comparison with state-of-the-art methods on RefCOCO [48], RefCOCO+ [48], and RefCOCOg [28] dataset task. We highlight the best and second best result obtained with the same backbone in bold and underlined.**

| Method | Backbone | RefCOCO | | | RefCOCO+ | | | RefCOCOg | | | ReferIt |
|---|---|---|---|---|---|---|---|---|---|---|---|
| | | val | testA | testB | val | testA | testB | val-g | val-u | test-u | test |
| **Two-stage** | | | | | | | | | | | |
| VC [49] | VGG16 | - | 73.33 | 67.44 | - | 58.40 | 53.18 | 62.30 | - | - | - |
| MAttNet [47] | ResNet-101 | 76.65 | 81.14 | 69.99 | 65.33 | 71.62 | 56.02 | - | 66.58 | 67.27 | 29.04 |
| RvG-Tree [12] | ResNet-101 | 75.06 | 78.61 | 69.85 | 63.51 | 67.45 | 56.66 | - | 66.95 | 66.51 | - |
| CM-Att-Erase [25] | ResNet-101 | 78.35 | 83.14 | 71.32 | 68.09 | 73.65 | 58.03 | - | 67.99 | 68.67 | - |
| NMTree [24] | ResNet-101 | 76.41 | 81.21 | 70.09 | 66.46 | 72.02 | 57.52 | 64.62 | 65.87 | 66.44 | - |
| Ref-NMS [3] | ResNet-101 | 80.70 | 84.00 | 76.04 | 68.25 | 73.68 | 59.42 | - | 70.55 | 70.62 | |
| **One-stage** | | | | | | | | | | | |
| ReSC-Large[42] | DarkNet-53 | 77.63 | 80.45 | 72.30 | 63.59 | 68.36 | 56.81 | 63.12 | 67.30 | 67.20 | 64.60 |
| SAFF [45] | DarkNet-53 | 79.26 | 81.09 | 76.55 | 64.43 | 68.46 | 58.43 | - | 68.94 | 68.91 | - |
| TransVG [6] | ResNet-50 | 80.32 | 82.67 | 78.12 | 63.50 | 68.15 | 55.63 | 66.56 | 67.66 | 67.44 | 69.76 |
| D-MDETR [32] | ResNet-50 | 81.62 | 83.85 | 76.24 | 67.00 | 70.95 | 58.13 | 68.04 | 70.14 | 69.57 | 71.13 |
| LADS [33] | ResNet-50 | 82.85 | 86.67 | 78.57 | 71.16 | 77.64 | 59.82 | - | 71.56 | 71.66 | 71.08 |
| HFRN [30] | ResNet-101 | 79.76 | 83.12 | 75.51 | 66.80 | 72.53 | 59.09 | - | 69.71 | 69.08 | - |
| TransVG [6] | ResNet-101 | 81.02 | 82.72 | 78.35 | 64.82 | 70.70 | 56.94 | 67.02 | 68.67 | 67.73 | 70.73 |
| LGFPN [36] | ResNet-101 | 81.76 | 84.78 | 78.16 | 70.29 | 76.19 | 59.68 | 69.20 | 73.06 | 73.24 | **73.61** |
| LUNA [22] | ResNet-101 | 84.67 | 86.74 | 80.21 | 72.79 | 77.98 | **64.61** | - | 74.16 | 72.85 | 72.97 |
| **Ours** | | | | | | | | | | | |
| MMCA$_{TransVG}$ | ResNet-50 | **84.34** | **86.99** | **80.06** | **72.44** | **78.01** | **63.86** | **72.02** | **74.11** | **73.46** | **72.87** |
| MMCA$_{TransVG}$ | ResNet-101 | **84.76** | **87.34** | **80.86** | **73.18** | **78.67** | 64.13 | **72.53** | **74.91** | **73.87** | 73.43 |

convolution block and apply our method on the last bottleneck block in the last three layers (C3, C4, and C5 layers).

## 4 EXPERIMENTS

### 4.1 Datasets

**RefCOCO/ RefCOCO+/ RefCOCOg.** RefCOCO [48] includes 19,994 images with 50,000 referred objects. The samples in Ref-COCO are officially split into a train set with 120,624 expressions, a validation set with 10,834 expressions, a testA set with 5,657 expressions and a testB set with 5,095 expressions. Similarly, Ref-COCO+ [48] contains 19,992 images with 49,856 referred objects and 141,564 referring expressions. It is also officially split into a train set with 120,191 expressions, a validation set with 10,758 expressions, a testA set with 5,726 expressions and a testB set with 4,889 expressions. RefCOCOg [28] has 25,799 images with 49,856 referred objects and expressions. There are two commonly used split protocols for this dataset. One is RefCOCOg-google [28], and the other is RefCOCOg-umd [29]. We report our performance on both RefCOCOg-google (val-g) and RefCOCOg-umd (val-u and test-u) to make comprehensive comparisons.

**ReferItGame.** ReferItGame [18] includes 20,000 images collected from the SAIAPR-12 dataset [8]. We follow the same split as in the previous works [6, 46] to divide this dataset into three subsets and report our results.

### 4.2 Implementation Details

Our experiments are mainly based on TransVG [6], QRNet [46] and VLTVG [40]. For the MMCA$_{TransVG}$ and MMCA$_{VLTVG}$, the visual branch employ the ResNet-50 as its CNN-based backbone,

followed by 6 transformer encoder layers, where the embedding dimension is set as 256, the head number of multi-head attention modules is set as 8, and the hidden dimension in FFN is set as 2048, aligning with the configuration in TransVG and VLTVG. Additionally, since TransVG does not provide the Swin-S [27] based model, the results and implementation of TransVG (Swin-S) we adopted follow the QRNet, and our MMCA$_{TransVG (Swin-S)}$ uses the same architecture. We employ the basic BERT [7] for textual feature generation and follow the TransVG to process the input images and sentences. We also follow the training setting used in TransVG, QRNet and VLTVG, which use AdamW optimizer with weight decay $10^{-4}$. The batch size is set to 64 and the learning rate is set to $10^{-5}$ for pre-trained parameters and $10^{-4}$ for other parameters. The parameters without pretraining are randomly initialized with Xavier. We train MMCA$_{TransVG}$, MMCA$_{VLTVG}$ for 90 epochs and MMCA$_{TransVG (Swin-S)}$, MMCA$_{QRNet (Swin-S)}$ for 160 epochs, which is consistent with previous works [6, 46] for fair comparison. The learning rate is multiplied by a factor of 0.1 at epoch 60. The hyperparameters $k$ and $C_{mm}$ in the gate network are set to 4 and 128. We also follow the data augmentation strategies employed in previous works [6, 40, 42, 43, 46].

### 4.3 Comparisons with State-of-the-art Methods

In Table 1, we compare our proposed model with other state-of-the-art methods on RefCOCO [48], RefCOCO+ [48], and RefCOCOg [28] datasets. For small-sized models, which use ResNet-50 as CNN backbone, our method based on TransVG has better performance with +4.02% / +4.32%/ +1.94% on RefCOCO, +8.94%/ +9.86%/ +8.23% on RefCOCO+, and +5.46%/ +6.45% +6.02% on RefCOCOg, which outperforms the recent methods and achieve the state-of-the-art

**Table 2: Results with stronger baseline.**

| Method | Backbone | RefCOCOg | | ReferIt |
|---|---|---|---|---|
| | | val-u | test-u | test |
| VLTVG [40] | ResNet-50 | 74.90 | 73.88 | 71.60 |
| TransVG [46] | Swin-S | 69.34 | 68.99 | 70.86 |
| QRNet [46] | Swin-S | 73.03 | 72.52 | 74.61 |
| VG-LAW [34] | Swin-S | 75.61 | **76.28** | 74.83 |
| **Ours** | | | | |
| MMCA$_{VLTVG}$ | ResNet-50 | 75.48 | 75.11 | 73.89 |
| MMCA$_{TransVG}$ | Swin-S | 73.58 | 73.59 | 74.11 |
| MMCA$_{QRNet}$ | Swin-S | **76.08** | 75.64 | **75.86** |

**Table 3: Ablative experiments on the weight type.**

| Weight type | RefCOCO | | |
|---|---|---|---|
| | val | testA | testB |
| $W_m$ | 81.67 | 83.99 | 78.34 |
| $W_c$ | 82.98 | 85.14 | 78.99 |
| $W_v$ | 82.14 | 84.12 | 78.01 |
| $W_q, W_k$ | 83.35 | 85.89 | 79.58 |
| $W_q, W_k, W_v$ | 82.96 | 84.81 | 78.22 |
| $W_q, W_k, W_c$ | **84.34** | **86.99** | **80.06** |
| $W_q, W_k, W_c, W_m$ | 83.69 | 86.24 | 79.71 |

**Table 4: Ablative experiments with different modal information as inputs. T: Textual features, V: Visual features, (A): Add fusion, (G): Gated fusion.**

| Modality | RefCOCO | | |
|---|---|---|---|
| | val | testA | testB |
| - | 80.32 | 82.67 | 78.12 |
| T | 83.54 | 85.55 | 79.74 |
| V | 82.21 | 84.92 | 78.27 |
| T+V (A) | 83.34 | 86.06 | 79.11 |
| T+V (G) | **84.34** | **86.99** | **80.06** |

**Table 5: Ablative experiments on the hyperparameter rank $r$**

| Rank $r$ | RefCOCO | | | params (M) |
|---|---|---|---|---|
| | val | testA | testB | |
| - | 80.32 | 82.67 | 78.12 | 149.52 |
| 4 | 82.87 | 84.71 | 79.17 | 151.17 |
| 8 | 83.36 | 85.64 | 79.12 | 151.34 |
| 16 | 83.65 | 86.28 | 79.51 | 151.69 |
| 32 | 83.91 | 86.51 | 79.67 | 152.40 |
| 64 | **84.34** | **86.99** | **80.06** | 153.80 |

results. When the larger visual backbone (ResNet-101) adopted, our method still gain overall better performance on all four datasets. Compared with the the most recent works, our method generally surpasses LGFPN [36] and LUNA [22] on the RefCOCO and Ref-COCOg datasets.

Table 2 also reports the performance of our method based on the stronger baseline. We compare our method with the VLTVG [40], TransVG (Swin-S) [46], QRNet [46] and VG-LAW [34] on the RefCOCOg and ReferItGame datasets. It is noted our method enhances visual grounding models by guiding the behavior of the visual encoder and does not introduce a new model structure. And we intentionally avoid comparing models with different structures. The results indicate that our method can still achieve consistent improvement under different strong baselines. Although VG-LAW has not made its source code available, our MMCA$_{QRNet}$ still outperforms it with +0.47% and +1.03% on the RefCOCOg val split and ReferItGame dataset.

### 4.4 Ablation Studies

In this section, we perform ablation studies on the RefCOCO [48] dataset to assess the effectiveness of our proposed method. We establish TransVG (ResNet-50) as the baseline due to its straightforward and CNN-transformer mixed architecture. Our analysis centers on three key aspects: where to add these adaptations, the effectiveness of gated fusion and comparison with different rank $r$ .
**Where to Apply the Adaptations.** In Table 3, we discussed which weights our proposed MMCA should apply to the network. We applied our method to different parts of the self-attention layer $W_q, W_k, W_v$, convolutional layer $W_c$, and FFN layer $W_m$ with the

rank $r = 64$. The results on the validation and testing sets of RefCOCO show that applying our mehod on the $W_q, W_k, W_c$ have significantly higher performance than others. In the self-attention layer, using MMCA in $W_q, W_k$ yields better results than in $W_v$ or $W_q, W_k, W_v$, this could be because attention score plays a role in feature selection, making the network pay more attention to text-relevant regions.
**Effectiveness of Gated Fusion.** To verify the effectiveness of gated fusion of multimodal features in MMCA, we use different modal information to guide weight update matrices and present the experimental results in Table 4. It can be seen that fusing multimodal features will bring better results than using only textual or visual features. To further verify the impact of the gating mechanism, we adopt a simple summation method, e.g. $E_{mm} = E_t + E_v$, to fuse visual and textual features and compared it with the fusion method using the gating mechanism. The results show that the gating scheme can further bring improvement with 1.00%, 0.93% and 0.95%. This also verifies that our method can effectively dynamically control the input of textual information to handle more complex scenarios.
**Comparison with Different Rank $r$.** At last, we analyze the effectiveness of rank $r$. We investigate the number of rank $r$ in MMCA, for $W_q, W_k, C$, by comparing the detection performance and the number of parameters on RefCOCO. As shown in Table 5, our proposed method already performs well with a very small rank $r = 8$. With 1.82M additional parameters, our method achieved 3.04%, 2.57%, and 1.00% improvement on dataset val, testA, and testB. It can be seen that the overall performance gradually increases as rank $r$ gets larger. And we take the best $r = 64$ as our default setting for other ablation study and state-of-the-art result.

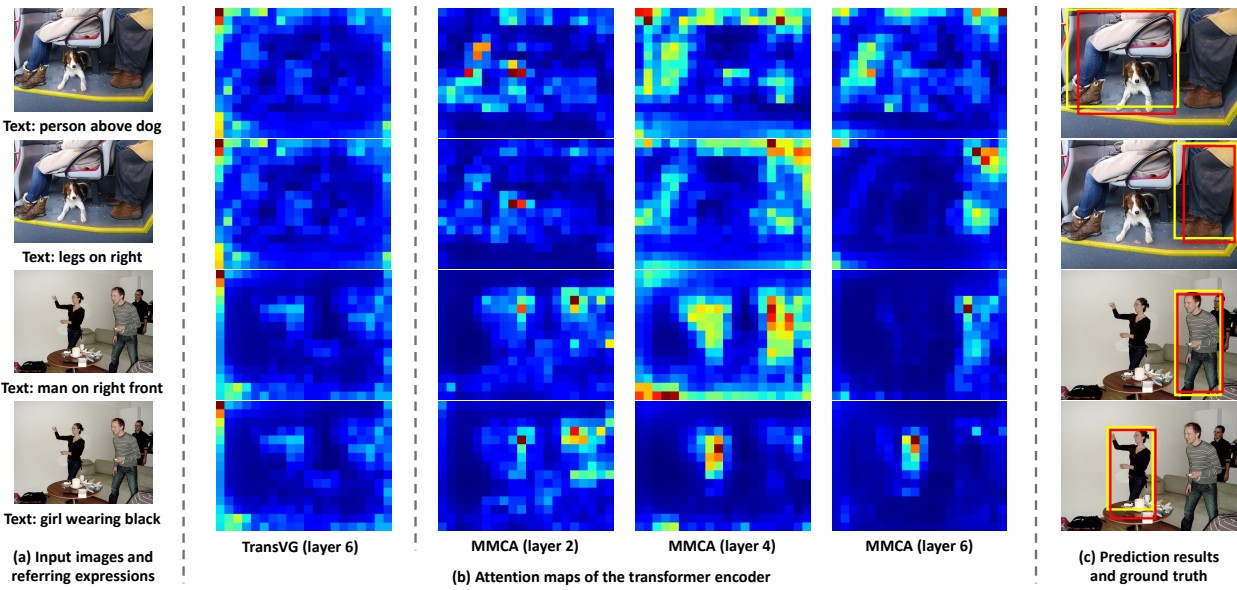

**(a) Input images and referring expressions**

Text: person above dog

Text: legs on right

Text: man on right front

Text: girl wearing black

TransVG (layer 6)

MMCA (layer 2)

MMCA (layer 4)

MMCA (layer 6)

**(b) Attention maps of the transformer encoder**

**(c) Prediction results and ground truth**

**Figure 5: Visualization of input images and referring expressions, the attention maps of the transformer encoder layer in TransVG and MMCA, our prediction results (red bounding boxes) and ground truth (yellow bounding boxes).**

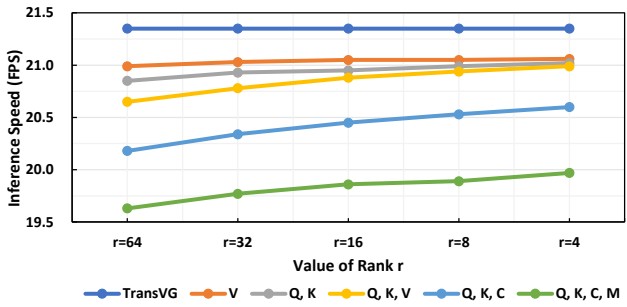

**Figure 6: The inference speed (FPS) of our method with different factors rank $r$ and weight type, where Q, K, V, C, M denotes the weight type $W_q, W_k, W_v, W_c, W_m$.**

## 4.5 Efficiency Analysis

We examined the inference time of our proposed method with various factor dimensions by setting $r = \{4, 8, 16, 32, 64\}$ and applying our method to different weight matrices of TransVG. The experiments were conducted on a single NVIDIA RTX3090 GPU and the results are presented in Figure 6. The experimental results suggest that our method shows robust inference efficiency across different hyperparameter values $r$. We observed that the primary inference delay arises from the adaptation applied in the FFN layer and convolutional layer. We attribute this delay to the larger input channels and the presence of additional branches and we believe that this can be improved through more detailed parameter settings, such as additional channel reduction for convolutional or fully connected layers. When comparing our best result, achieved by adopting rank $r = 64$ and applying it to $W_q, W_k, C$, to the baseline, we observed a decrease in inference speed of approximately 5%

(1.07 FPS drop). When considering the trade-off between accuracy and inference speed, we recommend applying MMCA to $W_q$ and $W_k$, which can achieve 83.35%, 85.89%, 79.58% on RefCOCO while only brings about 0.48 FPS drop.

## 4.6 Qualitative Results

In Figure 5, we visualize the input images, referring expressions, multi-modal conditional visual encoder's attention maps, final prediction results and ground truth. It can be observed that the attention scores are generally higher on the foreground regions or regions relevant to the corresponding expression. The comparison with TransVG shows the ability of our proposed MMCA to focus on the object regions. And as the number of encoder layers deepens, the attention distribution gradually concentrates from the general foreground area to the object referred to in textual expressions, which validates the effectiveness of our method.

## 5 CONCLUSION

In this paper, we propose Mulit-modal Conditional Adaption (MMCA) to address the limitation of independent visual feature extraction for visual grounding. MMCA integrate visual and textual information to reorganize a set of low-rank weight matrices and enable the visual encoder can adaptively update its weight to concentrate on the text-relevant regions. Extensive experiments and ablation studies have validated the high effectiveness of our method. Our proposed framework significantly outperforms the baseline and achieves comparable results with the state-of-the-art methods while little parameter budget and time cost required. In future work, we plan to introduce this idea into the parameter-efficient tuning of large-scale multi-modal model and explore the mechanism behind conditional adaption, e.g. how are the conditional weight update enable the visual model to extract expression-relevant visual feature.

## ACKNOWLEDGMENTS

This work was in part supported by the Hainan Provincial Joint Project of Sanya Yazhou Bay Science and Technology City (Grant No. 2021JJLH0099), the National Key Research and Development Program of China (Grant No. 2022ZD0160604), the National Natural Science Foundation of China (Grant No. 62176194), the Young Scientists Fund of the National Natural Science Foundation of China (Grant No. 62306219). the Key Research and Development Program of Hubei Province (Grant No. 2023BAB083), and the Project of Sanya Yazhou Bay Science and Technology City (Grant No. SCKJ-JYRC-2022-76, SKJC-2022-PTDX-031).

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
