# OpenReview forum: "Visual Grounding with Multi-modal Conditional Adaptation"
_acmmm.org/ACMMM/2024/Conference — MM2024 Oral_

### Official Review · Reviewer_g9Ef · 2024-05-11

**Rating:** 5
**Confidence:** 3

**Summary:**

This paper focuses on the task of visual grounding. Considering that the current method requires sophisticated designs to mitigate the poor performance of the model that uses the independent visual encoder, this paper proposes the multi-modal conditional adaptation method which enables the visual encoder to update its weights adaptively and focus on the text-relevant regions. Concretely, the proposed method utilizes a set of weighting coefficients to reorganize the weight update matrices to update the visual encoder. Extensive experiments are conducted on the benchmark datasets.

**Strengths:**

- The proposed multi-modal conditional adaptation approach innovatively updates the weights of the visual encoder to specifically enhance focus on text-relevant regions.
- This paper is well organized and easy to read.
- The proposed approach achieved state-of-the-art performance on the benchmark datasets.

**Limitations:**

- The Flikr30k Entities dataset is also widely used in Visual Grounding in previous works, such as TransVG, and QRNet. It would be better to see the robustness of the proposed method in this large-scale dataset.
- Some important references are missing. For example, the reference to LoRA in line 162.

**Suitability:**

3

---

### Official Review · Reviewer_1FFa · 2024-05-24

**Rating:** 4
**Confidence:** 3

**Summary:**

This paper introduces a novel approach to visual grounding called Multi-modal Conditional Adaptation (MMCA), which enhances the feature extraction process of visual encoders in visual grounding tasks. The MMCA method integrates visual and textual information to create multi-modal embeddings and uses these embeddings to dynamically update the weights of the visual encoder, allowing it to focus more on text-relevant regions. Extensive experiments on four widely used datasets demonstrate that MMCA significantly improves performance and achieves state-of-the-art results. Ablation studies further highlight the method's lightweight and efficient nature.

**Strengths:**

1、This paper proposes a novel Multi-modal Conditional Adaptation (MMCA) method that addresses feature extraction challenges in visual grounding, which achieves significant performance enhancements on visual grounding tasks across four datasets.

2、Extensive experiments were conducted to validate the performance of the proposed approach.

3、The paper has a well-organized structure, and the explanation of the algorithm design is relatively clear.

**Limitations:**

1、In the performance comparison, the different methods do not use consistent backbones, which may compromise the fairness of the evaluation.

2、The paper does not include a discussion on future research directions, which leaves readers uncertain about the potential for further development and exploration of this work.

Questions:

In Table 4, using T alone appears to perform better than using T+A. What might be the reason for this?

**Suitability:**

2

---

### Official Review · Reviewer_Npf9 · 2024-05-25

**Rating:** 5
**Confidence:** 2

**Summary:**

The paper titled "Visual Grounding with Multi-modal Conditional Adaptation" addresses the challenge of visual grounding, which is the task of locating objects in images as specified by natural language expressions. The authors propose a novel method called Multi-modal Conditional Adaptation (MMCA) to enhance the visual encoder's ability to focus on text-relevant regions within an image. The key innovation of MMCA is the adaptive updating of the visual encoder's weights using information from both visual and textual modalities. This approach is intended to overcome the limitations of existing methods that use independent visual encoders, which can struggle to differentiate between multiple objects within the same image described by different text. The paper reports significant improvements over baseline models and state-of-the-art results on four widely used datasets, demonstrating the effectiveness and efficiency of the proposed method.

**Strengths:**

1. The paper introduces a new method, MMCA, which is innovative in its approach to visual grounding by adaptively updating the weights of the visual encoder based on multi-modal information. This represents a novel contribution to the field.
2. The method is well-defined, and the authors provide a clear explanation of how MMCA works, including the generation of multimodal embeddings and the application of weighting coefficients.
3. The paper includes extensive experiments on four datasets, which is adequate for evaluating the proposed method. The results are compared against state-of-the-art methods, adding credibility to the claims.
4. The paper is well-organized, and the authors have made efforts to explain their method clearly, including providing visualizations of attention maps and prediction results. And there are abundant ablation implementations to demonstrate the reliability of the method.

**Limitations:**

1. While the method is described as efficient, the paper could provide more details on the computational complexity and whether the adaptive weight updating process could become computationally expensive for larger models or datasets.
2. The method shows improvements on the evaluated datasets, but it is not clear how well MMCA would perform on other types of images or with other languages, which could be a limitation in terms of generalizability.
3. The paper does not mention code release or detailed training procedures, which are important for reproducibility.

**Suitability:**

3

---

### Official Review · Reviewer_yrfZ · 2024-05-28

**Rating:** 4
**Confidence:** 3

**Summary:**

The author proposed multi-modal conditional adaptation(MMCA), a method that can enable the visual encoder to directly focus on text-relevant regions. The MMCA enables the networks to combine textual features and visual features, and then use a gated fusion module to avoid overfitting on the training data. The author employed MMCA in different visual grounding architectures and showed convincing experiment results. Ablation studies show the effectiveness of the proposed modules.

**Strengths:**

1. The paper is well-organized and easy to follow, the intuition of the idea is discussed in the introduction.
2. The proposed MMCA is technically sound and explained clearly in detail.
3. The proposed MMCA is a plug-in module and can be used in different visual grounding architectures.
4.  The experimental results are comprehensive and convincing.

**Limitations:**

1. In the table.3, is there any analysis to explain why adding W_m performs worse?
2. In the table.5, looks like with larger rank r will help the method improve the performance, the author should keep increasing the rank and see where the bottleneck is.
3. Could you provide the experimental result in the Flicker30K dataset?

**Suitability:**

3

---

### Meta-Review · Area_Chair_xZto · 2024-07-03

**Recommendation:** Accept (Oral)
**Confidence:** 5

**Metareview:**

This paper proposes a Mulit-modal Conditional Adaption (MMCA) method to alleviate the limitation of independent visual feature extraction for visual grounding. MMCA enables the visual encoder adaptively update its weight to concentrate on the text-relevant regions. Extensive experiments and ablation studies validate the effectiveness of the proposed method.

(+) On the positive side, the reviewers found the method to be interesting and effective and appreciate the clear motivation and impressive results.

(-) On the negative side, there are still some concerns on the insufficient discussion, missing references and clarity of the writing.

Several technical questions and suggestions were raised by the reviewers. The authors have taken these into consideration. Overall, all reviewer agreed to accept the paper after the rebuttal. Therefore, the AC recommends accepting the paper.